# Neurologic Consultations and Headache during Pregnancy and in Puerperium: A Retrospective Chart Review

**DOI:** 10.3390/jcm12062204

**Published:** 2023-03-13

**Authors:** Julia S. M. Zimmermann, Mathias Fousse, Ingolf Juhasz-Böss, Julia C. Radosa, Erich-Franz Solomayer, Ruben Mühl-Benninghaus

**Affiliations:** 1Department of Gynecology, Obstetrics and Reproductive Medicine, University Medical School of Saarland, 66421 Homburg/Saar, Germany; 2Department of Neurology, University Medical School of Saarland, 66421 Homburg/Saar, Germany; 3Department of Gynecology, University Medical Center Freiburg, 79106 Freiburg, Germany; 4Department of Diagnostic and Interventional Neuroradiology, University Medical School of Saarland, 66421 Homburg/Saar, Germany

**Keywords:** cranial magnetic resonance imaging, headache, puerperium, pregnancy

## Abstract

Headache is a common symptom during pregnancy and in puerperium that requires careful consideration, as it may be caused by a life-threatening condition. Headaches in pregnant women and women in puerperium are classified as primary or secondary; acute, severe and newly diagnosed headaches should prompt further investigation. We aimed to further characterise the demographic features, symptoms, examination findings, and neuroimaging results of cases of headache during pregnancy and in puerperium. All pregnant women or women in postpartum conditions who attended neurological consultations at the emergency department of the clinic for Gynaecology, Obstetrics and Reproductive Medicine of Saarland University/Germany between 2001/2015 and 2012/2019 were enrolled in this retrospective chart review. Data collected from the charts included demographic/pregnancy characteristics, clinical features and imaging findings. Descriptive statistics as well as binary logistic regression were performed. More than 50% of 97 patients had abnormal findings in their neurological examination. Magnetic resonance imaging findings were pathological for almost 20% of patients—indicating conditions such as cerebral venous thrombosis, reversible posterior leukoencephalopathy, brain tumour and intracranial bleeding. The odds of abnormal neuroimaging results were 2.2-times greater among women with abnormal neurological examination findings than among those with normal examination results. In cases of headache during pregnancy and in puerperium, neuroimaging should be indicated early on. Further research is needed to determine which conditions indicate a need for immediate neuroimaging.

## 1. Introduction

Headache is a common symptom during pregnancy and in puerperium, and it requires careful consideration as a life-threatening condition could underlie it. The reported prevalence of headache in pregnancy is as high as 35% [1]. The occurrence of headache in general, and migraines in particular, is more likely in women than in men due to the differences in their oestrogen levels [2,3]. In addition, secondary causes of headache such as hypertensive disorders, cerebrovascular disorders with transient global amnesia, space-occupying lesions and infections are more likely to occur during pregnancy due to alterations in the maternal physiology [4,5,6]. As in all patients, headaches in pregnant women and those in puerperium should be classified as primary or secondary according to the International Classification of Headache Disorders 3 criteria of the International Headache Society (IHS) [7]. Acute, severe or newly diagnosed headache in pregnancy should always prompt further investigation [8]. Most headaches are not associated with intracranial lesions, but some are the first symptoms of severe pathologies—including cerebral venous thrombosis, intracranial bleeding, stroke, tumour, autoimmune encephalopathies and eclampsia [9,10].

Inpatient neurological consultations for pregnant women and women in puerperium due to headache are common [11]. Emergent evaluations of complaints of headache require the rational selection of immediate neuroimaging examinations, but guidelines on whether and when magnetic resonance imaging (MRI) is indicated are lacking [12,13]. Some clinicians base the indication for a neuroimaging study on abnormal findings during neurological examinations, whereas others always perform MRI in such cases. Thus, we conducted this study to better characterise the demographic features, symptoms, examination findings and neuroimaging results of cases from neurological consultations for headache in pregnant women and women in puerperium.

## 2. Materials and Methods

### 2.1. Study Design and Patients

For this retrospective chart review, all cases in which pregnant women and women in puerperium attended neurological consultations at the emergency department of the clinic for Gynaecology, Obstetrics and Reproductive Medicine of Saarland University, Homburg, Germany were considered. 

All cases in which neurologic consultations were requested for the chief complaint of headache and other neurological symptoms between January 2015 and December 2019, and in which the women underwent cranial and—in some cases—additional spinal MRI, were screened for inclusion. Additional inclusion criteria were defined as follows: 

Women being pregnant or in puerperium aged ≥ 16 years with complete data in their charts concerning the target variables, presenting with acute headache and/or additional neurologic symptoms, met the inclusion criteria. Puerperium or the postpartum period were defined as the first six weeks following childbirth. A cranial MRI must have been performed as a diagnostic tool to meet the inclusion criteria. 

The exclusion criteria were defined as follows: patients having undergone a CT scan, patients not having undergone cranial MRI and women who were not pregnant or were more than six weeks postpartum. Patients with incomplete data in their charts concerning the target variables were excluded, as well as patients aged younger than 16 years.

Data collected from the charts included demographic characteristics (age), pregnancy characteristics (i.e., gravidity, parity and gestational age), history of neurological conditions, clinical features (i.e., pain location, laterality, duration and associated symptoms including nausea, vomiting, seizure, visual aura, vision disorder (defined as visual disturbances during a headache) and sensibility disorder) and imaging findings. Normal and abnormal neurological examination findings such as mental status, vigilance disturbances, cranial nerve status, sensory and motor impairments and gait and coordination, were also recorded. Abnormal neuroimaging findings were classified as incidental or pathological based on the diagnosis of the primary care-providing neuroradiologist. Headaches were classified according to the International Classification of Headache Disorders 3 criteria of the International Headache Society (IHS) as primary (accompanied by normal neuroimaging findings) or secondary (accompanied by abnormal neuroimaging findings) [7]. Nevertheless, incidental findings could be found by MRI in patients with primary headaches.

According to the local ethics committee regulations (Saarland institutional review board), ethical approval was not required for this retrospective review of medical findings.

### 2.2. Statistical Analysis

The data were collected in a Microsoft Excel (version 16.35; Microsoft Corporation, Redmond, WA, USA) database free of patient identifiers. For the calculation of descriptive statistics, data were transferred to SPSS (version 19; SPSS Inc., Chicago, IL, USA). Initially, the normality of data distributions was determined using the Kolmogorov–Smirnov test. Categorical data are reported as frequencies with percentages; data are expressed as medians and ranges for continuous variables. Binary logistic regression was used to determine the likelihood that patients with focal neurological examination findings had pathological intracranial conditions. Independent statistical significance for all analyses was defined as a two-sided *p* value < 0.05. 

## 3. Results

In 311 cases, neurologic consultations and cranial MRI were requested for a chief complaint of headache or other neurological symptoms between January 2015 and December 2019 at the emergency department of the clinic for Gynaecology, Obstetrics and Reproductive Medicine of Saarland University, Homburg, Germany. Considering the inclusion and exclusion criteria described above, 97 patients were considered for the final analysis.

According to a review of the patient charts, the median age of the 97 women included in the review was 31 years (range 16–49 years), the median numbers of pregnancies and previous live births were two (range 1–7) and one (range 0–5). At the time of consultation, the median gestational age was 33 weeks (2–41) and 23 (23.7%) women were in puerperium (Table 1, Figure 1). A history of headache was present in 12.4% of patients (Table 1).

The primary care-providing gynaecologist requested formal neurological consultations for all patients. The chief complaint of headache—present in 54 (55.7%) cases—was frequently accompanied by seizure (n = 13 (13.4%)), vision disorder (n = 21 (21.6%)), paraesthesia (n = 12 (12.4%)), vertigo or dizziness (n = 11 (11.3%)), syncope (n = 6 (6.2%)), aphasia (n = 6 (6.2%)), hypertensive events (n = 9 (9.3%)) and nausea or vomiting (n = 6 (6.2%); Table 1). Abnormal neurological findings were present in 52 (53.6%) patients; 6 (6.2%) of the 97 patients refused further examination and the remaining 91 (93.8%) underwent additional neuroimaging examinations. Emergent neuroimaging consisted of cranial and, in some cases, additional spinal MRIs. Taking a closer look at the cranial pathologies, we excluded another three (3.1%) patients who only underwent spinal imaging from the definite analysis of pathologies. 

Cranial MRI revealed underlying headache causes in 19 (19.6%) cases. MRI findings were normal in 63 (64.9%) cases and incidental in six (6.2%) cases. Incidental findings included lesions in the medullary cavity, pituitary prominence as a physiological change occurring during pregnancy, and nonspecific punctate foci of hyperintensity in the white matter. Pathological intracranial conditions detected in this patient sample were cerebral venous thrombosis (n = 2 (2.1%)), posterior reversible encephalopathy syndrome caused by eclampsia (n = 7 (7.2%)), intracranial haemorrhage (subdural haematoma; n = 1 (1%)), cerebrovascular accident/stroke (n = 2 (2.1%)), aneurysm (n = 1 (1%)), sinusitis (n = 1 (1%)), brain tumour (astrocytoma; n = 1 (1%)), cerebrospinal fluid leak syndrome (CSFLS); n = 1 (1%)), cerebral parenchymal lesion (n = 1 (1%)), cyst of Rathke’s pouch (n = 1 (1%)) and vascular dissection (n = 1 (1%); Figure 2, Figure 3 and Figure 4).

Thirteen (25%) patients with abnormal neurological findings also had pathological neuroimaging findings (posterior reversible encephalopathy syndrome, n = 6 (11.5%); aneurysm, n = 1 (1.9%); cerebral venous thrombosis, n = 3 (5.8%); cerebrovascular accident, n = 3 (5.8%); astrocytoma, n = 1 (1.9%); cerebral parenchymal lesion, n = 1 (1.9%)). Thus, 39 (75%) of those women with pathologies in neurologic examinations had normal results in MR imaging. 

Eleven (57.9%) of the 19 women with abnormal neuroimaging findings (median age, 30 years, range: 19–40 years) were pregnant, with a median gestational age of 33 weeks (19–41 weeks, range); all of these women were in the second or third trimester (Table 1). The remaining eight (42.1%) women were in the postpartum period. Two (10.5%) women with abnormal neuroimaging findings (vascular dissection and cerebrospinal fluid leak syndrome, respectively) had histories of migraine (Table 1).

In the secondary headache group, hypertensive disorders of pregnancy and in puerperium were the most frequently assigned diagnoses. All of the seven women with posterior reversible encephalopathy syndrome (three (42.9%) of whom were pregnant and four (57.1%) of whom were in the postpartum period) had hypertensive events, elevated uric acid levels and D-dimers; six (85.7%) of them had seizures and two (28.6%) had elevated blood levels when checking the HELLP parameters (hemolysis, elevated liver enzymes, low platelet count). Only one (14.3%) of these patients had no focal finding upon neurological examination. One of the two women with headache secondary to cerebral venous thrombosis presented at the emergency department with facial paresis, and the other woman with this condition had an elevated D-dimer. The woman diagnosed with an aneurysm of the arteria cerebri media made the chief complaints of headache and hemi-paraesthesia. The woman diagnosed with vascular dissection of the vertebral artery had acute severe headache in childbed. Another diagnosis found by MRI was a subdural hematoma, becoming manifest in acute headache in childbed. The two women who had cerebrovascular accidents had vision disorder, and one of these women had the chief complaint of headache. The patient with a newly diagnosed astrocytoma had vision disorder and headache for several days. Seizures occurred in six (n = 31.6%) patients with secondary headache and seven (9%) of those with normal neuroimaging findings. Two patients who had seizures and no eclampsia had previously known epilepsy.

Neurological examination findings were more likely to be abnormal in patients with secondary headache (50% in the primary headache group vs. 68.4% in the secondary headache group). The odds of abnormal neuroimaging results were 2.2-times greater among women with abnormal neurological examination findings than among those with normal examination results, but this difference was not significant (*p* = 0.15; 95% confidence interval, 0.75–6.28; Figure 5).

## 4. Discussion

Among the cases included in this chart review, 19.6% of pregnant women and women in childbed presented to the emergency room with headache whose underlying etiology was revealed by neuroimaging; this percentage is similar to those reported previously and seems to be representative of the overall population [14,15]. The most frequent causes of secondary headaches during pregnancy and in puerperium were PRES caused by preeclampsia (7.2%), cerebral venous thrombosis (2.1%), intracranial haemorrhage (1%) and vascular dissection (1%); these numbers are in line with a review of the literature, reporting numbers from 3.2–6% for PRES, from 1.4–6% for cerebral venous thrombosis, from 1.4–3 for intracranial haemorrhage and 1.6% for vascular dissection [14,15,16].

Primary headache is diagnosed by exclusion; in the absence of appropriate diagnostic evaluation—such as by MRI—any new and/or severe headache occurring during pregnancy must be considered to be a symptom of an underlying disease. Primary headaches (e.g., migraine, tension headache, trigeminal autonomic cephalalgia and cluster headache) occurring during pregnancy normally resolve in the second or third trimester; only 10% of women describe a worsening of symptoms [17,18]. Secondary headache should always be considered—and the primary care-providing physician should be alerted—when a patient has no history of headache, when neurological examination findings are abnormal and/or in the presence of any of the following: hypertensive disorder, worsening of a known headache, changes in a headache with postural changes, headaches that awaken the patient, headaches caused by physical activity, thrombophilia, fever, trauma, seizure and history of acute infection or tumours—these features should prompt immediate MRI examination [17]. 

As a further caution, known migraines should not prompt an automatic diagnosis of primary headache. In this study, neuroimaging revealed additional abnormalities in two women with migraine histories; despite the presence of formerly known migraines, their current headaches had other causes. However, in those two cases, a red flag was present: their headaches differed from their baseline headache. 

Nevertheless, physicians should refrain from performing MRI for every pregnant patient presenting with headache, as those unfamiliar with the actual risks of this modality may be unnecessarily concerned about its effects. Neurological examination is very useful in this regard, but should not be the only basis for decisions concerning indications for neuroimaging. The majority of patients with secondary headache in this study certainly had abnormal neurological examination findings, and the odds that neuroimaging would reveal a pathological intracranial condition in these patients were 2.2-times greater—albeit not statistically significant. However, 50% had abnormalities in the neurologic examination and no pathologic findings in MRI. A referable neurological examination abnormality did not independently predict secondary headache in this study. Several algorithms have been developed for the evaluation of patients with headache [14,19,20,21]: First, the physician should identify any “red flags”/warning signs in the patient’s history or physical examination findings. In the absence of such signs, the headache should be treated as primary. If it does not improve over time, additional diagnostic work-up should be considered. In the presence of such signs, intoxication and a history of preeclampsia should be ruled out, and MRI should be performed. In the absence of any abnormal neuroimaging findings, additional laboratory evaluations, lumbar puncture and/or examinations with another neuroimaging modality should be conducted [21].

Alterations of the maternal physiology over the course of pregnancy increase the risk of several dangerous secondary headache disorders associated with vascular endothelial dysfunction and hypertensive disorders of pregnancy [4]. This factor could explain the occurrence of secondary headaches exclusively in the second and third trimesters among pregnant women in our sample. The majority of women with pathological neuroimaging findings in this study were in puerperium, during which time (at least in the first few days) the maternal physiology remains altered. Moreover, some cerebrovascular disorders—such as cerebral venous thrombosis—appear to occur more frequently in the postpartum period, as delivery induces vascular changes and endothelial dysfunction [16]. Thus, women in the postpartum period should receive the same care as do pregnant women.

The limitations of the present study include its retrospective design. In addition, outpatient cases of typical preeclampsia and those treated with first-line acute therapies are not referred routinely for neurological consultation in our institution, which likely biased our sample toward more atypical and severe cases. Thus, prospective studies are required to better capture the overall population of pregnant women and women in puerperium presenting with headache. Finally, data on headache characteristics were missing from many emergency department records examined in this study, which prevented us from further classifying headaches according to the IHS guidelines.

## 5. Conclusions

The indication for neuroimaging examinations should be evaluated early in cases of headache in pregnant women and women in puerperium, as an underlying headache aetiology frequently exists. In German hospitals, MRI is the preferred modality in such cases, as it avoids radiation exposure and the introduction of contrast agents which threaten the foetus. Further research is needed to determine which clinical factors are critical for decisions concerning neuroimaging indications in this context. No clinical feature has been identified yet as being predictive of the presence of a pathological lesion in acute neuroimaging studies.

## Figures and Tables

**Figure 1 jcm-12-02204-f001:**
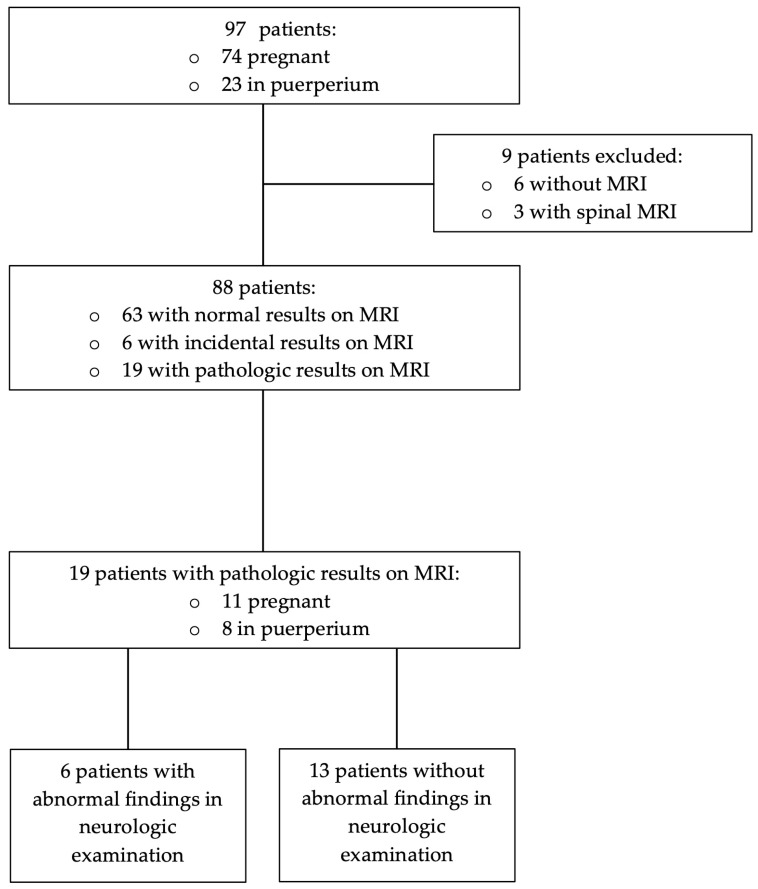
Study population. A total population of 97 patients from the emergency department of the clinic for Gynaecology, Obstetrics and Reproductive Medicine of Saarland University, Homburg, Germany between January 2015 and December 2019 was included.

**Figure 2 jcm-12-02204-f002:**
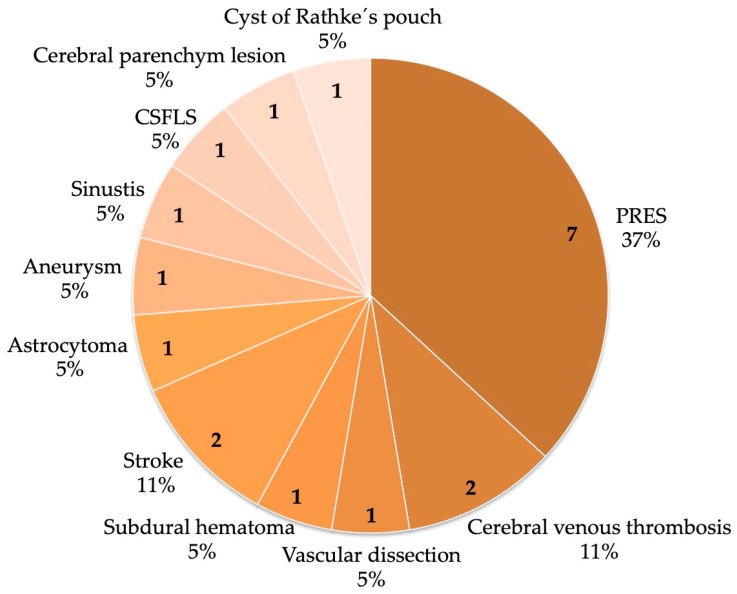
Abnormal findings in neuroimaging PRES = posterior reversible encephalopathy syndrome, CSFLS = cerebrospinal fluid leak syndrome. The absolute values are given in bold within the pie; the percent values refer to the 19 patients.

**Figure 3 jcm-12-02204-f003:**
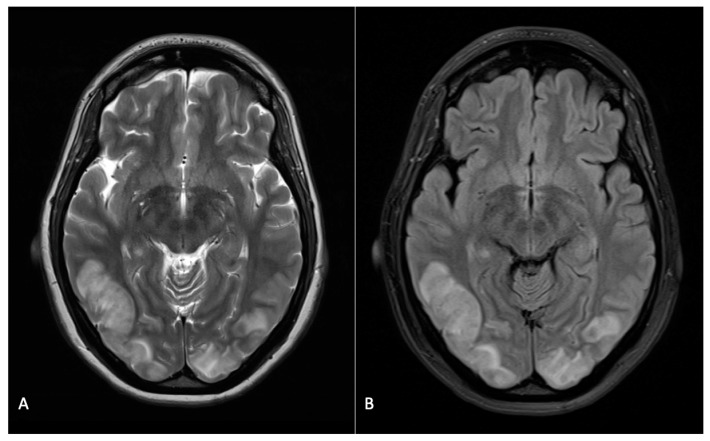
Representative MR images for PRES. Pregnant woman with posterior progressive headache due to PRES. MRI reveals bilateral occipital hypertensities of the brain parenchyma and cortex on T2 (**A**) and FLAIR (**B**) images.

**Figure 4 jcm-12-02204-f004:**
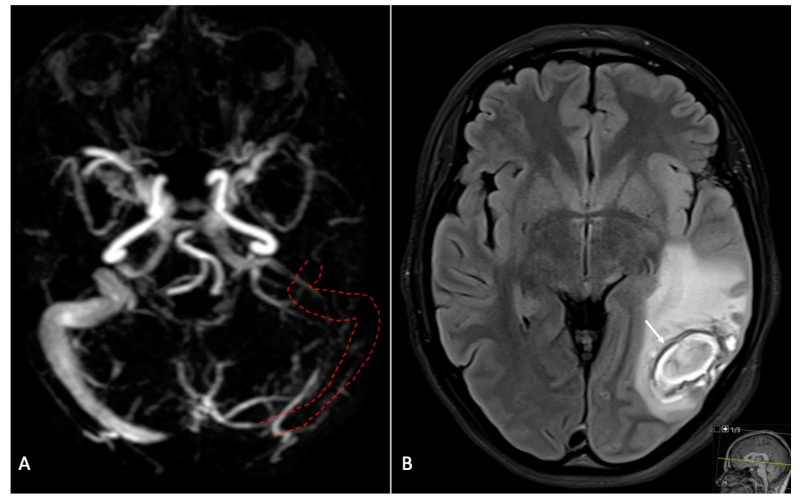
Representative MR images for sinus thrombosis. Pregnant woman with progressive left-sided headache due to sinus and vein thrombosis. Note the absence of the left transverse and sigmoid sinus (red dashed lines) on the MR angiography (**A**) causing parenchymal bleeding of the left temporal lobe (arrow), visible on the FLAIR image (**B**).

**Figure 5 jcm-12-02204-f005:**
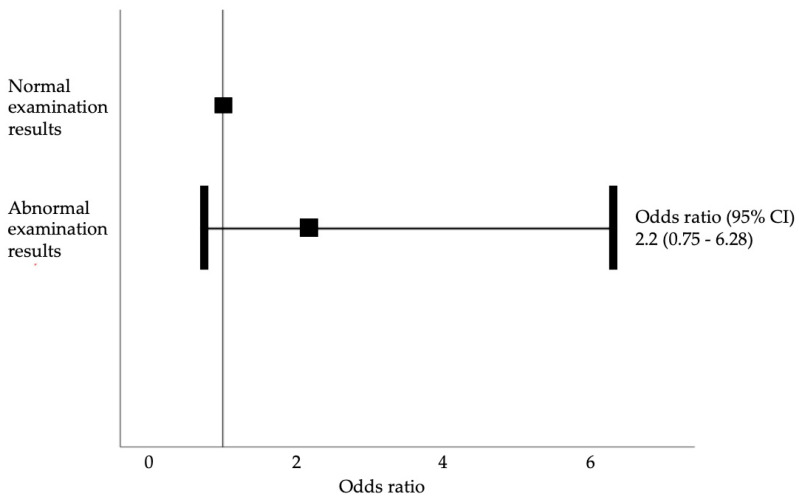
Odds ratio of abnormal neuroimaging results in neurological examination (*p* = 0.15), representing the likelihood of neuroimaging changes in relation to the normal/abnormal neurological examination.

**Table 1 jcm-12-02204-t001:** Patients’ characteristics.

Characteristic	All Headaches (*n* = 97)	Secondary Headaches (*n* = 19)
Age (median, range)	31 years (16–49 years)	30 years (19–40 years)
Gestations (median, range)	2 (1–7)	2 (1–5)
Deliveries (median, range)	1 (0–5)	1 (0–5)
Gestational age (median, range)	33 weeks (2–41 weeks)	33 weeks (19–41 weeks)
Pregnant women (*n*,%)	74 (76.3%)	11 (57.9%)
Women in puerperium (*n*, %)	23 (23.7%)	8 (42.1%)
History of migraine (*n*, %)	12 (12.4%)	2 (10.5%)
**Abnormalities in neurologic examination (*n*,%)**	52 (53.6%)	13 (68.4%)
**Pyramidal**		
Hyperreflexia	1 (1.1%)	0 (0%)
**Extrapyramidal**		
Rigor	1 (1.1%)	0 (0%)
**Cerebellar**		
Vertigo/dizziness	11 (11.3%)	0 (0%)
Nausea	6 (6.2%)	0 (0%)
**Other symptoms**		
Behavioral changes	2 (2.1%)	1 (5.3%)
Somnolence	2 (2.1%)	0 (0%)
Syncope	6 (6.2%)	0 (0%)
Paresis	2 (2.1%)	1 (5.3%)
Facial nerve paresis	4 (4.1%)	1 (5.3%)
Aphasia	6 (6.2%)	2 (10.5%)
Seizure	13 (13.4%)	6 (31.6%)
Vision disorder	21 (21.6%)	5 (26.3%)
Paresthesia	12 (12.4%)	1 (5.3%)
Hypertensive events	9 (9.3%)	7 (36.8%)

## Data Availability

The dataset used and analyzed during the current study is available from the corresponding author on reasonable request.

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
