# Peer review of "Neurologic Consultations and Headache during Pregnancy and in Puerperium: A Retrospective Chart Review"

_jcm, 2023, doi:10.3390/jcm12062204_

Round 1
Reviewer 1 Report
The review of the literature lacks statistical data on the various causes of headache during pregnancy.
Most of the references are more than 5 years old.
Criteria of inclusion and exclusion must be described in detail
97 patients were included in the study of medical charts of patients with headache between January 2015 and December 2019. How many patient charts were studied prior to exclusion to obtain a final 97 patients?
Statistics should be obtained in comparison with the entire original number of patients, and not only with MRI patients. May be after that percentages of received data will be lower.
Author Response
Comments to reviewer 1:
The review of the literature lacks statistical data on the various causes of headache during pregnancy.
Thank you for this valuable advice, we added statistical data on the various causes of headache in pregnancy in the discussion (line 291-296).
Most of the references are more than 5 years old.
We added some more recent literature (for details please see reference section).
Criteria of inclusion and exclusion must be described in detail
Thank you for this remark, we revised the methods section as required (line 67-79)
97 patients were included in the study of medical charts of patients with headache between January 2015 and December 2019. How many patient charts were studied prior to exclusion to obtain a final 97 patients?
Thank you for this valuable advice, the number of charts studied prior to exclusion was 311 and was added to the results section (line 118-123).
Statistics should be obtained in comparison with the entire original number of patients, and not only with MRI patients. May be after that percentages of received data will be lower.
Thank you for this remark, tough we did not obtain statistics in comparison with the original number of patients, as in this big data base were also non-pregnant women and non-pregnant women suffering from cancer. Besides our data obtained were in line with findings of the literature.
Reviewer 2 Report
The paper submitted to me for review concerns neurologic consultations and headache during pregnancy and in puerperium and it's a retrospective paper from before the COVID-19 pandemic. The analysis concerns 97 women. However interesting and important the topic of the paper is, it contains a number of shortcomings and gaps that need to be filled:
1. Every sentence of the Results section begins with "the median ..." - this should be rephrased.
2. What are "vision disorders"? (line 140) are they ophthalmic diseases diagnosed by an ophthalmologist or vision/visual disturbances/pfenomena during a headache? visual aura?
3. line 151 uses the phrase "non specific punctate loci..." no such term is used to refer to radiological changes in patients with headaches
4. what is meant by abnormal neurological examination? there is no table or summary of abnormalities on neurological examination - were there pyramidal, extrapyramidal, cerebellar symptoms?
5. the paper is about headaches and there is no summary of the characteristics of headaches themselves, it is not known whether there was a history of headaches, whether they increased during pregnancy? what types of headaches were found in patients?
6. neurological symptoms in pregnancy also need to be differentiated from transient global amnesia (based on https://pubmed.ncbi.nlm.nih.gov/32643675/) and autoimmune encephalopathies (based on https://pubmed.ncbi.nlm.nih.gov/36139446/) - this information should be supplemented in the introduction
Author Response
comments to reviewer 2:
Reviewer 2
The paper submitted to me for review concerns neurologic consultations and headache during pregnancy and in puerperium and it's a retrospective paper from before the COVID-19 pandemic. The analysis concerns 97 women. However interesting and important the topic of the paper is, it contains a number of shortcomings and gaps that need to be filled:
- Every sentence of the Results section begins with "the median ..." - this should be rephrased.
Thank you for this valuable remark, we rephrased it as required (line 124-128).
- What are "vision disorders"? (line 140) are they ophthalmic diseases diagnosed by an ophthalmologist or vision/visual disturbances/pfenomena during a headache? visual aura?
Thank you for this valuable remark, we explained the term in the methods section (line 83-84).
- line 151 uses the phrase "non specific punctate loci..." no such term is used to refer to radiological changes in patients with headaches
There was a misunderstanding, the text refers to punctate foci (new line 166), which is standard in radiology (https://www.reference.com/science-technology/punctate-foci-7004332028e187fa).
- what is meant by abnormal neurological examination? there is no table or summary of abnormalities on neurological examination - were there pyramidal, extrapyramidal, cerebellar symptoms?
Thank you, we defined the term abnormal neurological examination in the methods section (line 84-87). We added the abnormalities on neurological examination to Table 1 and classified them as pyramidal, extrapyramidal and cerebellar.
- the paper is about headaches and there is no summary of the characteristics of headaches themselves, it is not known whether there was a history of headaches, whether they increased during pregnancy? what types of headaches were found in patients?
Thank you for this valuable remark. We added the history of headaches and the accompanying clinical features to Table 1. As the data in the charts are lacking detailed information on the development during pregnancy, we could not comment on this detail. The different types of headaches in the secondary headache group with the characteristics and partial development of headaches are described in the text from line 258-276.
- neurological symptoms in pregnancy also need to be differentiated from transient global amnesia (based on https://pubmed.ncbi.nlm.nih.gov/32643675/) and autoimmune encephalopathies (based on https://pubmed.ncbi.nlm.nih.gov/36139446/) - this information should be supplemented in the introduction
The introduction was complemented with the additional information.
Reviewer 3 Report
This brief report deals with a topic of headache in pregnancy and puerperium and aims to better characterise the demographic features, symptoms, examination findings and neuroimaging results in cases of neurological consultation for headache. I think it is necessary to emphasize the difference between ‘new onset headache’ and ‘pre-existent headache’, specifying different features for each patient.
- Table 1 -‘Patients’ characteristics’- is incomplete, missing the clinical information listed in section ‘materials and methods-study design and patients’
- Figure 1: the drawing of the figure is not clear and the terms used are not suitable (i.e. headache cause, normal neurology). The caption is confusing, you need to choose numbers or letters
- Figure 2: it would be preferable to insert also the absolute value. in the caption, “list of different headache causes (n=19)' is unnecessary
- Line 186: ‘progredient’ should be replaced with ‘progressive’
- Statistical analysis and results should be listed in a table or figure to be more comprehensible
Author Response
comments to reviewer 3:
This brief report deals with a topic of headache in pregnancy and puerperium and aims to better characterise the demographic features, symptoms, examination findings and neuroimaging results in cases of neurological consultation for headache. I think it is necessary to emphasize the difference between ‘new onset headache’ and ‘pre-existent headache’, specifying different features for each patient.
- Table 1 -‘Patients’ characteristics’- is incomplete, missing the clinical information listed in section ‘materials and methods-study design and patients’
Thank you, we revised Table 1 as required.
- Figure 1: the drawing of the figure is not clear and the terms used are not suitable (i.e. headache cause, normal neurology). The caption is confusing, you need to choose numbers or letters
Thank you for this valuable advice, we changed the figure and caption as required.
- Figure 2: it would be preferable to insert also the absolute value. in the caption, “list of different headache causes (n=19)' is unnecessary
Thank you for this valuable advice, we changed it as required.
- Line 186: ‘progredient’ should be replaced with ‘progressive’
Thank you for this valuable advice, the term was changed as required.
- Statistical analysis and results should be listed in a table or figure to be more comprehensible
We added Figure 5 for better comprehension of the statistical analysis and results.
Round 2
Reviewer 1 Report
Wonderful, very interesting work.
Fit the record under figure № 1.
Delete one of the figure № 2 (duplicate!!).
Author Response
Thank you very much, please excuse these errors. We changed formatting as requested.
Reviewer 2 Report
The authors have addressed my comments in a thorough and comprehensive manner and all have been included in the revision. Thank you. The paper is ready for printing.
Author Response
Thank you very much.
Reviewer 3 Report
- Figure 5: the caption is not explaining the figure; it represents the likelihood
of neuroimaging changes in relation to the normal/abnormal neurological examination
- Line 293: what do you mean by "red flags"?
Author Response
Thank you for this advice, we changed the caption of Figure 5 as requested.
Apart from that we added an explanation to the term ‘red flags’.